# WLAN Aware Cognitive Medium Access Control Protocol for IoT Applications

**Asfund Ausaf** [1]**, Mohammad Zubair Khan** [2,*]**, Muhammad Awais Javed** [1,*] **and Ali Kashif Bashir** [3]

[1] Department of Electrical and Computer Engineering, COMSATS University Islamabad, Islamabad 45550, Pakistan; asfundausaf@gmail.com
[2] Department of Computer Science, College of Computer Science and Engineering, Taibah University Madinah, Madina 42353, Saudi Arabia
[3] Department of Computing and Mathematics, Manchester Metropolitan University, Manchester M15 6BH, UK; dr.alikashif.b@ieee.org
[*] Correspondence: mkhanb@taibahu.edu.sa (M.Z.K.); awais.javed@comsats.edu.pk (M.A.J.); Tel.: +966-537562956 (M.Z.K.); +92-3355927096 (M.A.J.)

**Abstract:** Internet of Things (IoT)-based devices consist of wireless sensor nodes that are battery-powered; thus, energy efficiency is a major issue. IEEE 802.15.4-compliant IoT devices operate in the unlicensed Industrial, Scientific, and Medical (ISM) band of 2.4 GHz and are subject to interference caused by high-powered IEEE 802.11-compliant Wireless Local Area Network (WLAN) users. This interference causes frequent packet drop and energy loss for IoT users. In this work, we propose a WLAN Aware Cognitive Medium Access Control (WAC-MAC) protocol for IoT users that uses techniques, such as energy detection based sensing, adaptive wake-up scheduling, and adaptive backoff, to reduce interference with the WSN and improve network lifetime of the IoT users. Results show that the proposed WAC-MAC achieves a higher packet reception rate and reduces the energy consumption of IoT nodes.

**Keywords:** wireless sensor networks; medium access control; IEEE 802.15.4; IEEE 802.11

## 1. Introduction

Wi-Fi is one of the major contributors to today's global connectivity and economy [1]. With a significant increase in wireless data transfer demands in recent years, the wireless spectrum has become a scarce and expensive resource. As the number of Internet of Things (IoT) devices utilizing the open spectrum bands (like the 2.4 GHz band) is increased, this calls for a reevaluation of spectrum access protocols. Since the medium access control protocols are designed for a particular technology, they fail to achieve fair and efficient wireless resource sharing in the presence of interference from other miscellaneous technologies [2–9].

Wi-Fi is the major source of interference in the Industrial, Scientific, and Medical (ISM) radio band due to its high power and data rate as compared to the other technologies sharing the same 2.4 GHz ISM band [10]. Other technologies, such as IoT devices operating in the ISM band, can greatly benefit from the knowledge of current spectrum occupancy and design protocols for wireless coexistence, thus improving their transmission efficiency.

IoT devices consist of battery-powered wireless sensor nodes, and depletion of the battery will strip IoT devices from network communication. A common scenario in the future will be the co-existence of various IoT devices (consisting of IEEE 802.15.4 based wireless sensor nodes) with the already installed IEEE 802.11-based Wireless Local Area Network (WLAN). Both these wireless

technologies operate in the unlicensed ISM band of 2.4 GHz and cause interference to each others' transmissions. For this particular scenario, IoT devices will be subjected to more interference caused by the high-powered WLAN system. On the other hand, interference from the low-powered IEEE 802.15.4 transmissions will not impact the performance of the IEEE 802.11 devices [11].

A key issue in IoT is that WLAN nodes are unaware or blind to the existence of IEEE 802.15.4-based IoT devices due to the difference in the magnitude of transmission power. Therefore, WLAN nodes do not defer channel access when there is an ongoing packet transmission of packet IoT nodes. This overlapping of transmission results in frequent packet drop and energy loss for the IoT nodes when they are communicating with the IoT edge devices or sink nodes. An efficient medium access control protocol for IoT nodes is required to reduce packet loss due to transmission overlap so that the energy efficiency of IoT devices can be improved.

Performance evaluation of WLAN shows that its traffic is bursty, i.e., data frames are transmitted together in clusters and have short intervals between them, while the idle inactive periods between the cluster of frames are notably longer and can be exploited by IoT nodes for their transmission [12]. Cognitive medium access is a dynamic spectrum access method that allows secondary users to access the primary user's radio spectrum (which is not currently used by the primary user) [13–16].

In this paper, we propose a WLAN Aware Cognitive Medium Access Control (WAC-MAC) protocol that is adopted by the IEEE 802.15.4-compliant IoT nodes to reduce the interference of co-existing WLAN nodes. IoT nodes employ a sensing mechanism based on energy detection to find WLAN user activity and identify inactive periods of WLAN transmission. This is done by introducing a sensing phase in the superframe structure. WAC-MAC also uses an adaptive wake-up scheduling technique that reduces the current beacon interval in case the medium is busy after the sensing phase. This results in quicker sensing of the medium, again, instead of sleeping for the long beacon interval. Lastly, the WAC-MAC protocol uses an adaptive backoff scheme based on the node's battery lifetime. This mechanism allows nodes with less remaining energy to transmit quickly, hence conserving their energy. Results show that the proposed WAC-MAC protocol reduces the packet loss, as well as improves network energy life time and end-to-end delay. The following are the major contribution of our paper.

- Addition of a sensing phase in the superframe structure which allows IoT nodes to determine WLAN transmission and to subsequently identify extended inactive period between the cluster of WLAN frames.
- An adaptive wake-up scheduling scheme for IoT nodes which allows them to modify their superframe duration and beacon intervals in case the medium is busy with WLAN transmissions.
- An adaptive backoff scheme for IoT nodes which prioritizes the channel access for nodes with lower remaining battery life.

The rest of the paper is organized as follows. In Section 2, we provide a detailed review of the literature. Section 3 gives an overview and operation of the IEEE 802.11 and the IEEE 802.15.4 technologies. In Section 4, we provide the considered system model including interference, sensing, and energy consumption model. Section 5 describes the proposed WAC-MAC protocol. In Section 6, performance evaluation, including the simulation results, is presented. Finally, we conclude the paper in Section 7.

## 2. Related Work

Several ideas have been proposed to improve the communication and energy efficiency in IoT wireless sensor devices. In Reference [17], the authors minimize the idle listening time by turning off the radio of the sensor nodes when they are idle and not transmitting. This technique is implemented by using wake-up radios and an adaptive duty cycle control algorithm. In Reference [18,19], wireless sensor nodes tune to the best available communication band in a multi-channel network based on cross-network interference.

The authors in [20] propose techniques, such as frame aggregation and back-up channel availability, to improve the energy efficiency of the network. When both the sender and the receiver of a packet are the same, frame aggregation is performed at the MAC layer, where ten data frames are combined. Using this approach, packet overhead is reduced since packet headers are only attached once with the aggregated frame. To improve network reliability, backup channel availability is used in which, at any instant, if a primary user needs to access a channel when secondary users may be engaged in a communication, then without any transmission break-up, the secondary users move over to the available backup channel.

Authors in [21] propose a multi-channel MAC protocol in which secondary users reserve data channels using information exchange on a separate control channel. Nodes share information, such as channel sensing and destination node address, in the form of short preamble packets sent in series, rather than an extended preamble. Rather than being awake for the extended preamble, nodes that are not the destination for a packet go to sleep mode whenever the first short preamble is heard.

Reference [22] proposed a MAC protocol that improves channel utilization and energy efficiency. In a cognitive network where periodic spectrum sensing is performed, the transmission interval that is available for secondary users is often narrow which results in contention and reduced energy efficiency. The proposed scheme grants access to a single secondary user to transmit data packets before the start of the next spectrum sensing interval. Within this period, all other nodes go into their sleep mode, hence conserving the energy.

In Reference [23], authors propose a MAC protocol that mitigates WLAN interference experienced by the Zigbee devices using payload and header redundancy. WLAN transmissions can uniformly alter any of the bits in a Zigbee packet, thereby losing the whole frame. In the case of collisions, the header is generally the part that is altered. The proposed MAC protocol sends frames with multiple headers, allowing numerous opportunities to detect the packets. Having multiple headers increases network overhead, but it minimizes data loss, thereby reducing packet re-transmission, increasing network delivery rate, and improving energy consumption.

Authors in [24] propose a MAC protocol in which wireless sensors nodes (with packets to transmit) force WLAN to back off by sending a high powered jamming signal at the Distributed Interframe Spacing (DIFS) interval. The transmit power value of the jamming signal is set to the maximum WLAN power measured during the sensing period. The proposed MAC protocol reduces collision faced by the sensor nodes and results in a high packet delivery ratio. However, due to the use of jamming signals, the energy consumption of the network is increased.

A multi-hop cognitive MAC protocol is proposed in Reference [25] to reduce energy consumption caused by hidden WLAN terminals. Using continuous spectrum sensing, nodes decide about the available transmission opportunities. The next-hop distance for multi-hop communication is optimized based on the WLAN channel occupancy statistics. Results show an increased data throughput and reduced packet error rate.

A coexistence model between IEEE 802.15.4 and IEEE 802.11b/g is proposed by authors in [10]. The model uses factors, such as transmit power and transmission time, to establish co-existence ranges. The concept of co-existence ranges has been established by extending the concept of interference ranges and sensing across various wireless standards. Sensor nodes transmit packets by identifying IEEE 802.11 frame spaces and co-existence ranges.

Several works predict the availability of a free channel to avoid WLAN interference. For the case when there is a single WLAN access point with non-saturated traffic, Reference [26] modeled the arrival rate of WLAN packets as a Bernoulli process. A Poisson arrival process is assumed for WLAN traffic, and queuing technique (M/G/1) is used for output buffer modeling in Reference [27]. Inspired from the work of Reference [25,27], a superframe structure is proposed, with the added function of spectrum sensing to capture the two causes of inactivity; one is the time interval of the long white space, while the other is the short back off interval between two consecutive frames.

A major issue in the previous works is the high energy consumption of IoT nodes and inefficient utilization of WLAN extended inactive period. In this paper, we focus on these two problems and provide a solution.

## 3. Overview and Operation of IEEE 802.11 and IEEE 802.15.4

### 3.1. IEEE 802.11

IEEE 802.11 specifies medium access control, physical layer rules, and algorithms for WLAN. There are two modes of operation for implementation of Carrier Sense Multiple Access with Collision Avoidance (CSMA/CA) in WLAN that are supported by the IEEE 802.11: (i) Distributed Coordination Function (DCF): Contention-based access method that uses virtual carrier sensing along with physical sensing. Just like Ethernet, it first checks whether the radio link is free before transmitting. Stations use a random backoff after every frame to avoid collision, with the first transmitting devices seizing the channel. DCF may apply Clear To Send/Request To Send technique to further reduce the probability of collisions; and (ii) Point Coordination Function (PCF): Contention-free access method where the access point manages transmission between nodes using polling. To obtain priority over standard contention based utilities, PCF allows stations to transmit frames after a shorter interval [28].

WLAN DCF Traffic Model

WLAN traffic is bursty and its channel utilization is as shown in Figure 1. As can be seen, there exist gaps or periods of inactivity between two consecutive packets, as shown in Figure 2 [12]. WLAN data frames exist in clusters with short intervals between them, whereas the idle inactive periods between the clusters are longer. The reason for short intervals between the frames is due to the contention mechanism of the IEEE 802.11 MAC layer, in which senders take backoff for a short random interval before every transmission [29].

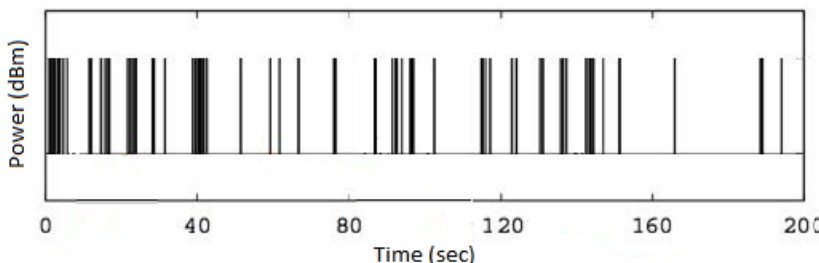

**Figure 1.** Traffic traces of Wireless Local Area Network (WLAN) networks in real-time envirnoment [12].

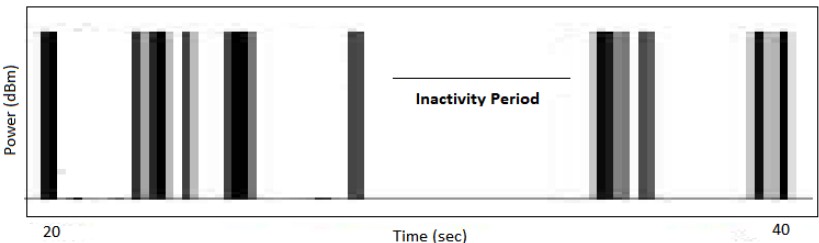

**Figure 2.** Enlarged view of WLAN traffic traces, packet transmission [12].

Most of the WLAN traffic uses Distributed Coordination Function (DCF) method. In case of DCF media access type, WLAN users sense the channel before initiating a transmission, as shown in Figure 3. If the medium is sensed as idle for Distributed Interframe Spacing (DIFS) time interval, the transmission takes place; otherwise, the node defers its transmission. After the DIFS interval, the node will generate a random backoff delay uniformly chosen between an interval $[0, W]$, which is

called the Contention Window (CW), where *W* is the size of the contention window. Initial *W* is set to a value of minimum contention window, or $CW_{min}$. Once the channel is again free for a DIFS time interval, the back-off timer is decremented. The backoff counter freezes if a transmission is detected on the medium. When the backoff timer reaches zero, the node transmits a data packet and waits for the Short Interframe Spacing (SIFS) interval before the next transmission [30].

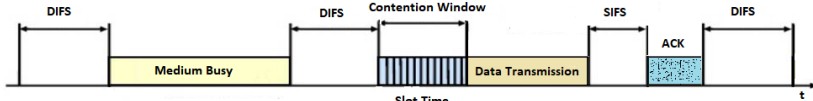

**Figure 3.** WLAN basic media access control scheme; slotted Carrier Sense Multiple Access with Collision Avoidance (CSMA/CA) contention-based channel access. DIFS = Distributed Interframe Spacing; SIFS = Short Interframe Spacing.

WLAN activity is divided into two states, the active state and the idle state. The active state consists of the time data is transmitted, including DIFS, SIFS, and Acknowledgment (ACK) intervals. The idle state consists of the WLAN contention period known as the contention window (CW) and WLAN inactive period known as white space (WS). Several works in the literature have modeled the active and inactive periods of WLAN [27,31]. As in Reference [25], active period $f_A(t)$ can be modeled by a uniform distribution, as given below:

$$f_A(t) = \frac{1}{\tau_{max} - \tau_{min}}, \ t \ \epsilon \ (\tau_{min}, \tau_{max}), \tag{1}$$

where parameters $(\tau_{max})$ and $(\tau_{min})$ define range of maximum and minimum values of the estimated active period.

Similarly, time spent by the WLAN users for contention, i.e., for backoff procedure, can be modeled by uniform distribution, as given below [25]:

$$f_{CW}(t) = \frac{1}{\tau_{BK} - 0}, \ t \ \epsilon \ (0, \tau_{BK}), \tag{2}$$

where parameters $(\tau_{BK})$ represent the maximum backoff time, given by the WLAN specification.

To estimate the idle duration due to WLAN inactive periods is more involved and depends on the type of user traffic. Several candidate distributions are widely used for spectrum activity modeling and are employed for deriving empirical IEEE 802.11 Cumulative Distribution Functions (CDF) of network idle-time [32]. In this paper, we employ Poisson distribution to estimate the WLAN inactive period with exponentially distributed inter-arrival time.

*3.2. IEEE 802.15.4*

The IEEE 802.15.4 is the MAC and physical layer standard for low rate wireless personal area networks. The IoT devices considered in this paper consist of IEEE 802.15.4-compliant sensor nodes and an infrastructure unit in the form of a network coordinator (sink node). IoT devices have low power and limited processing abilities that often communicate in single and multi-hop fashion. Each IoT sensor node consists of a sensing unit, storage unit, transceiver unit, processing unit, and power unit [33]. IEEE 802.15.4 employs two modes of operation. Beacon Mode, in which slotted Carrier Sense Multiple Access with Collision Avoidance (CSMA/CA), as shown in Figure 4, is used for data transmission. Clear Channel Assessment (CCA) is used in the physical layer to determine the channel occupancy [34]. CCA either performs Energy Detection or Carrier Sense, or a combination of both. CCA shall report a busy channel upon detection of any energy above the predefined energy threshold or a signal with known characteristics, such as modulation and spreading features. In IEEE 802.15.4

WSNs, a channel is sensed only during the CCA period. In *Non-Beacon Mode*, un-slotted CSMA/CA is used for data transmission [35].

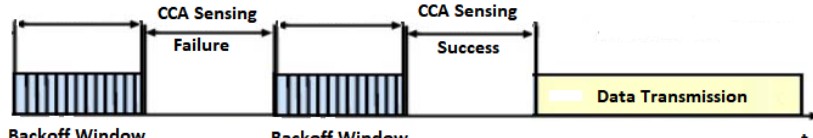

**Figure 4.** IEEE 802.15.4 basic media access control scheme; slotted CSMA/CA contention-based channel access. CCA = Clear Channel Assessment.

### 3.2.1. Beacon Mode Versus Non-Beacon Mode Comparison

Beacon-enabled mode has several advantages over non-beacon-enabled mode of sensor nodes. In beacon-enabled mode, regular beacons are sent by network coordinator (sink node) to synchronize all the sensor nodes and allocate Guaranteed Time Slot (GTS) for data transmission. Moreover, it supports a flexible Duty Cycle (DC), where the active time of the nodes can be adjusted as per the network scenario. In comparison, non-beacon enabled mode is suitable for an ad hoc communication scenario where there is no coordinator node.

### 3.2.2. IEEE 802.15.4 Superframe Structure

IEEE 802.15.4 uses a superframe structure, as shown in Figure 5, to control channel access and the network coordinator (sink node) transmits beacons at predetermined intervals [36]. The coordinator splits the superframe into active and inactive periods. The Beacon Interval (BI) is defined as the time interval after which the superframe is repeated. The Superframe Duration (SD) is defined as the time interval of the active period. The active period (or SD) is the sum of the Contention Access Period (CAP) and Contention Free Period (CFP). Each SD consists of 16 time slots, of which CFP is allocated 7 time slots. By using two parameters, namely the Beacon Order (BO) and the Superframe Order (SO), the coordinator controls the duty cycle of the superframe, i.e., BI and SD. During the CAP, the sensor nodes must contest with each other using slotted CSMA/CA to transmit the GTS request (for data transmission) to the coordinator. The coordinator then allocates GTS in the CFP of the next BI. In every BI, the nodes go into inactive mode to preserve energy [37].

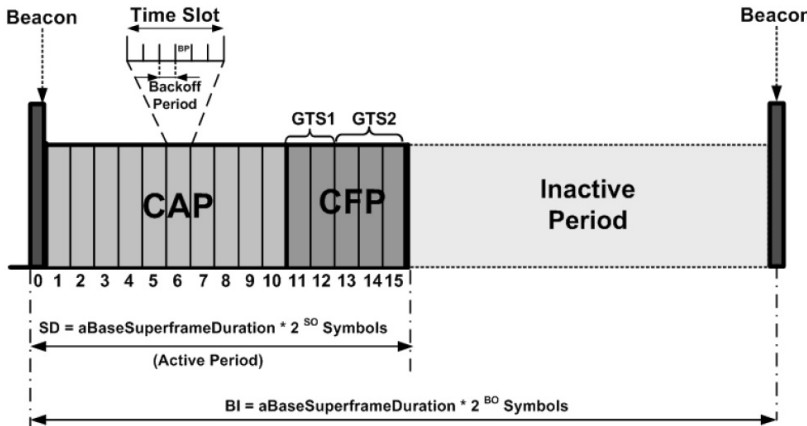

**Figure 5.** IEEE 802.15.4 superframe structure. CAP = Contention Access Period; CFP = Contention Free Period; GTS = Guaranteed Time Slot.

To evaluate Beacon Interval and Superframe Duration, the following equation can be used:

$$BI = 960 \times 2^{BO}, \tag{3}$$

$$SD = 960 \times 2^{SO}. \tag{4}$$

The Duty Cycle is given by the following equation:

$$DC = \frac{SO}{BO} = 2^{(SO-BO)}, \tag{5}$$

where $0 \leq SO \leq BO \leq 14$.

### 3.2.3. IEEE 802.15.4 Contention-Based Channel Access in Beacon-Enabled Mode

In the case of IEEE 802.15.4 contention-based channel access, the two main variables that control the channel access algorithm are the number of backoff (NB) and the backoff exponent (BE). Here, NB controls the number of times the CSMA/CA algorithm must take a backoff for channel access, and BE defines the number of backoff periods for channel access. At the start, NB is initialized with a value equal to zero and the value of BE is selected as $BE \; \epsilon \; (macMinBE, \; macMaxBE)$ with an initial value of $macMinBE$ (default value equal to 3). To avoid collisions, the contention algorithm uses a random wait time in the range of $[2^{BE} - 1]$. Here, one backoff unit period is equal to $20T_s$ ($T_s = 16$ µs). After the random backoff time, the channel is sensed for a time interval equal to Clear Channel Assessment (CCA). If the channel is busy, then the contention algorithm increments the values of BE and NB by one, while checking that BE does not increase beyond $macMaxBE$. If NB has a value of less than or equal to $macMaxBE$, the backoff process is repeated or the packet is dropped. If the channel is idle after CCA, CSMA/CA will immediately start the data transmission process [38].

## 4. System Model

The system model considered in this paper is shown in Figure 6. The IoT network includes a WLAN access point that covers the area in which IoT nodes are deployed. A beacon-enabled single-hop sensor network is considered, which consists of a network coordinator (sink node) or coordinator communicating with several IoT sensors $S = 1, 2, ..., N$. WLAN and IoT sensor nodes are static and operate at ISM Band (2.4 GHz). The transmission power of WLAN is usually around 12 dBm to $-20$ dBm. The WLAN stations are not aware of the IoT sensor nodes [29] as the WLAN carrier sense does not allow the detection of low-powered sensor node signals. This causes collisions and packet losses for the IoT nodes. On the other hand, the transmission power of the IoT sensor nodes is in the order of 0 dBm to $-3$ dBm [39], and their impact on WLAN transmissions is imperceptible [31]. To ensure efficient IoT communications, WLAN activity should be considered before transmission.

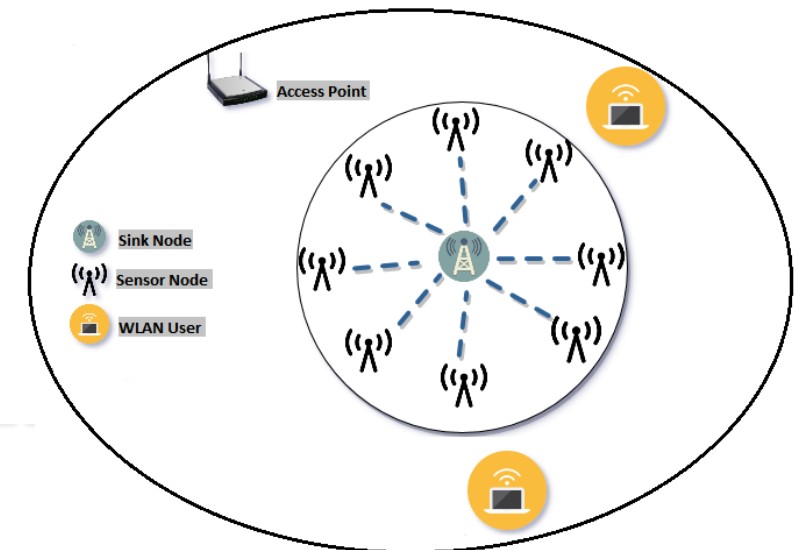

**Figure 6.** System model.

### 4.1. Interference Model

The signal propagation considered in this paper is based on the free space path loss model. For correct reception of a packet by the WSN node, the received power should be higher than the given threshold, denoted as $\lambda$, which is the Signal to Interference plus Noise Ratio (SINR) threshold. If a receiver IoT node comes within the transmission region of the WLAN transmitter, the packet of IoT node is lost. An interference radius $R_I$ (area from which the interference exists from a WLAN transmission) for IoT node can be expressed as follows [39]:

$$R_I = \sqrt[\eta]{\frac{(\lambda)(P_{WLAN})(P_{Lo})}{(P_{IoT})(P_{Lo})(r^{-\eta}) - (\lambda)(\sigma_N^2)}},$$ (6)

where $r$ is the distance between the transmitter and the receiver IoT sensor nodes, $\eta$ is the path loss exponent, $P_{IoT}$ is the IoT sensor node's transmit power, $P_{WLAN}$ is the WLAN transmit power, $P_{Lo}$ denotes the attenuation (at a distance of 1 m), and $\sigma_N^2$ is the noise power. In this research, we used a simple noise model where noise power is given as $N = kTB$. Here $k$ is Boltzmann constant, $T$ is temperature and $B$ is bandwidth. We ignored antenna noise temperature and receiver noise figure to simplify the noise model.

### 4.2. Sensing Model

IoT nodes carry out channel sensing based on energy detection through its incorporated Received Signal Strength Indicator (RSSI). Two types of sensing are performed by the sensor nodes; the first one is for identifying the WLAN inactive period, and the second one is for channel access control. Sensing measurement is based on sensor node's parameter called the *maximum sensitivity level ($\psi$)*. It is defined as the minimum signal SNR that can be detected by a sensor node. Sensing procedure within a finite time can be modeled as probabilistic energy detection, which is characterized by the parameters, such as the probability of missed detection $p_{MD}$ (signal is not detected) and the probability of false alarm $p_{FA}$ (when the sensing detects the signal while the channel is idle).

Energy detection of a signal depends on the estimation of energy decision threshold $\gamma$, which is a function of the probability of false alarm $p_{FA}$ and sensing time $t_s$ and is calculated for a certain value of the probability of false alarm. Probability of false alarm $p_{FA}$ is expressed by the following equation [40]:

$$p_{FA}(t_s; \gamma) = Q\left((\gamma - \sigma_N^2)/\sigma_N^2 \sqrt{2/(f_s t_s)}\right),$$ (7)

where $\sigma_N^2$ is noise power, $f_s$ is sampling frequency and $t_s$ is sensing time. Energy decision threshold $\gamma$ is given as follows:

$$\gamma(p_{FA}) = max\left\{\psi, \sigma_N^2\left[1 + \sqrt{(2/(f_s t_s)}Q^{-1}(p_{FA})\right]\right\}.$$ (8)

Sensor nodes determine the received signal power level and compare it with the energy decision threshold $\gamma$ to decide about the presence of the signal. Probability of missed detection $p_{MD}$ depends on the received signal power $P_{Rx}(d)$ and distance to the transmitter $d$:

$$P_{Rx}(d) = (P_{WLAN})(P_{Lo})(d^{-\eta}),$$ (9)

where $P_{WLAN}$ is WLAN transmit power, $P_{Lo}$ is attenuation at a reference distance, and $\eta$ is the path loss exponent. With known received signal power, we can determine the probability of missed detection, which is expressed as follows [40]:

$$p_{MD}(t_s, d; \gamma(p_{FA})) = 1 - Q\left(\frac{\gamma(p_{FA}) - (\sigma_N^2 + P_{Rx}(d))}{\sigma_N^2 \sqrt{(2/(f_s t_s)}}\right).$$ (10)

Complement of probability of missed detection is probability of detection, i.e., $(p_D = 1 - p_{MD})$, which is given as follows:

$$p_D(t_s, d; \gamma(p_{FA})) = Q\left(\frac{\gamma(p_{FA}) - (\sigma_N^2 + P_{Rx}(d))}{\sigma_N^2 \sqrt{(2/(f_s t_s)}}\right). \tag{11}$$

### 4.3. Energy Consumption Model

IoT sensor nodes operate in four modes; transmission, reception, sleep, and idle. The total energy consumed by the sensor node is the energy consumed during these four modes. Transmission mode is the one in which the sensor node transmits data to either the coordinator or other nodes. The total energy consumed by the sensor node in this phase depends on the size of the transmitted data [41]. In reception mode, the sensor node receives data from other nodes, such as the coordinator in our model, and the total energy consumed in this mode depends on the amount of data received by the sensor node. The idle mode is the one in which a node is not sending nor receiving packets, and energy is mainly spent to power the circuit. In sleep mode, the majority of the sensor circuitry is turned off, and the energy consumption of the sensor becomes minimal, depending on the sleep period duration. Thus, the total energy consumed by the sensor node is equal to

$$E_c = E_{tx} + E_{rx} + E_{sleep} + E_{idle}, \tag{12}$$

where $E_c$ is the total energy consumed during superframe, $E_{tx}$ is the energy consumed during data transmission, $E_{rx}$ is the energy consumed during data reception, $E_{sleep}$ is the energy consumed during sleep mode, and $E_{idle}$ is the energy consumed during idle periods.

$$E_{tx} = V \times I_{tx} \times t_{tx} = V \times I_{tx} \times \frac{L}{R}, \tag{13}$$

where $E_{tx}$ is the energy consumed in transmitting the data, $V$ is sensor battery voltage, $I_{tx}$ is the current consumed during the transmission, and $t_{tx}$ is the time required for transmission, which is equal to the ratio of packet size $L$ to the data rate $R$.

The reception mode energy consumption consists of energy consumed during the beacon received by the sensor nodes and given as:

$$E_{rx} = V \times I_{rx} \times t_{rx} = V \times I_{rx} \times \frac{Beacon}{R}, \tag{14}$$

where $I_{rx}$ is the current required during the reception, and $t_{rx}$ is the reception time.

Energy consumed during sleeping mode is given as:

$$E_{sleep} = V \times I_{sleep} \times t_{sleep}, \tag{15}$$

$$t_{sleep} = BI - SD = 2^{BO-SO}, \tag{16}$$

where $I_{sleep}$ is the current drawn during the sleep period. *BO* and *SO* are beacon order and superframe order values, respectively. Thus, the total energy consumed during a superframe is equal to:

$$E_c = V\left[(\frac{L}{R})(I_{tx}) + (2^{BO-SO})I_{sleep} + (\frac{Beacon}{R})(I_{tx})\right]. \tag{17}$$

### 4.4. Performance Analysis of Energy Detection

The performance of the energy detection procedure is evaluated in this portion by comparing simulation results with theoretical values. The energy detection technique relies on the energy of the received signal to detect primary users. The first step in energy detection is the filtering of the

out-of-band frequency signals. In the next step, these signals are passed through a square block and an adder block. If the resulting output is greater than a threshold $(\gamma)$, a licensed node is detected [42].

Received signal y(n) can be calculated from the following equation [43]:

$$y(n) = s(n) + w(n),\tag{18}$$

where $s(n)$ is the input signal power, $w(n)$ is the additive white Gaussian noise (AWGN) sample, and $n$ is the sample's index. Then, the decision metric based on the energy of the received signal can be given as

$$M = \frac{1}{N}\sum_{n=1}^{N}|y(n)|^2,\tag{19}$$

where $M$ is the decision metric, and $N = (f_s * t_s)$ is the number of samples. Comparing the decision metric $M$ to a threshold $\gamma$ corresponds to a selection between the following two hypotheses:

$$H_0 : y(n) = w(n),\tag{20}$$

$$H_1 : y(n) = s(n) + w(n),\tag{21}$$

where $H_0$ stands for detection of noise only, and $H_1$ stands for detection of noise and transmitted signal.

Receiver Operating Curve, or *ROC*, is often used to assess the performance of a detector based on two metrics: first is the probability of false alarm, and second is the probability of detection. ROC plots the probability of detection $P_D$ versus the probability of false alarm $P_{FA}$ for a given value of $SNR$.

Probability of detecting a transmitted signal is given as

$$P_D = P\left(M > \frac{\gamma}{H_1}\right).\tag{22}$$

Probability of false alarm can be defined as the detection probability of the frequency band being occupied when, in fact, there is no transmitted signal and can be written as

$$P_{FA} = P\left(M > \frac{\gamma}{H_0}\right).\tag{23}$$

As per the definitions of both probabilities, it is noticeable that, for probability of detection, high value is desired, and for probability of false alarm, low value is desired.

Figure 7 depicts the probability of detection of a WLAN user at different SNR values for a targeted value of probability of false alarm. $P_D$ increases with the increase in received $SNR$. The plot shows that the data obtained from simulation results closely match the theoretical values.

Figure 8 depicts the probability of detection of a WLAN user for different values of probability of false alarm. It can be seen that increasing $P_{FA}$ will result in increased chances of $P_D$.

Figure 9 depicts the probability of detection versus probability of false alarm for received $SNR = -12$ dB. Results show that there is a greater chance of false detection at higher $P_D$ values.

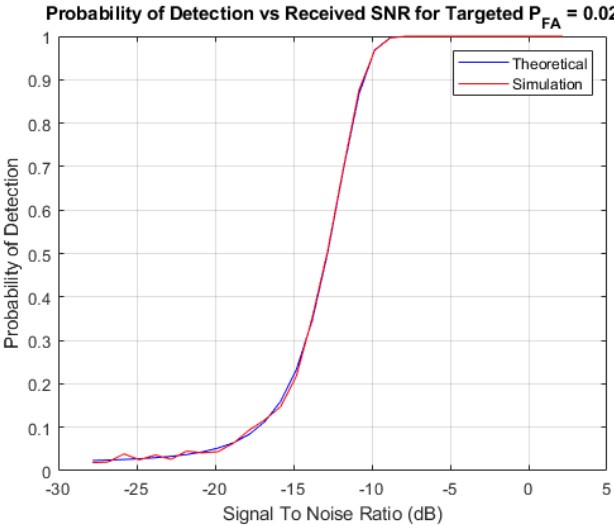

**Figure 7.** Receiver Operating Curve (ROC) plot for probability of detection vs received Signal To Noise Ratio (SNR).

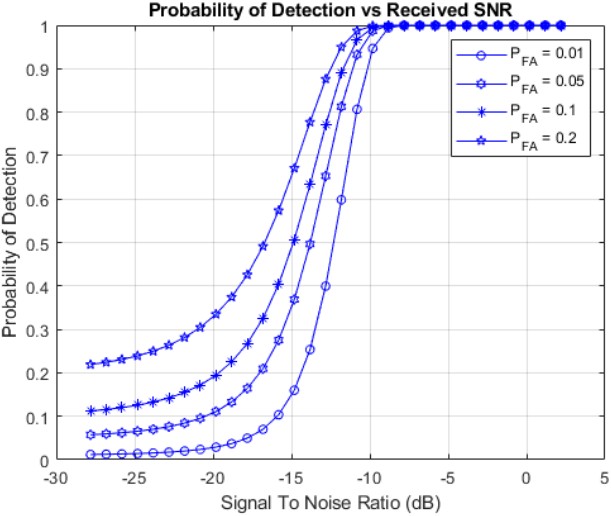

**Figure 8.** Plot for probability of detection vs SNR for different probability of false alarm.

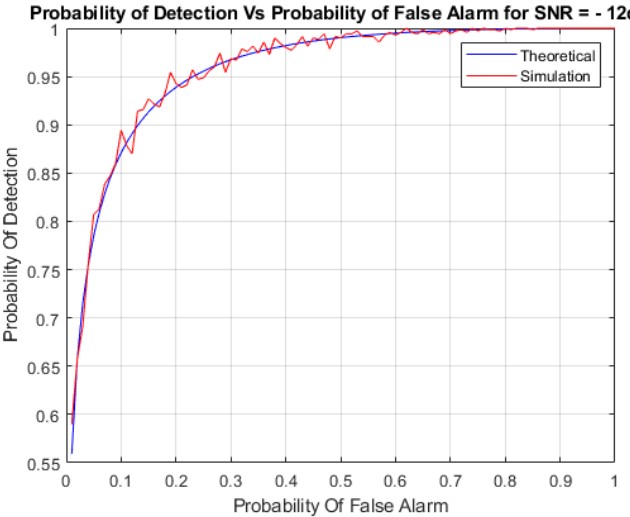

**Figure 9.** ROC plot for probability of detection vs probability of false alarm.

## 5. Proposed WLAN Aware Cognitive MAC (WAC- MAC) Protocol

In this section, we explain the proposed WLAN Aware Cognitive MAC (WAC-MAC) protocol for IoT nodes to improve energy efficiency and data throughput. WAC-MAC protocol modifies the superframe structure of IEEE 802.15.4 to incorporate sensing time slots for increased reliability. Moreover, an adaptive wake-up scheduling mechanism is introduced to improve the end-to-end delay of IoT data.

### 5.1. Sensing Time Slot in Superframe

WAC-MAC protocol adds a new time slot known as sensing time slot in the superframe structure, as shown in Figure 10. Each beacon interval starts with a beacon frame used by the coordinator for coordinator discovery and synchronization. Each sensor node then performs the spectrum sensing in the sensing time slot. The goal of spectrum sensing is to determine if the spectrum is idle or active using energy detection. The sensing time slot is greater than the maximum backoff value for WLAN transmissions. This is done to identify if the channel is empty due to WLAN contention or extended inactive period.

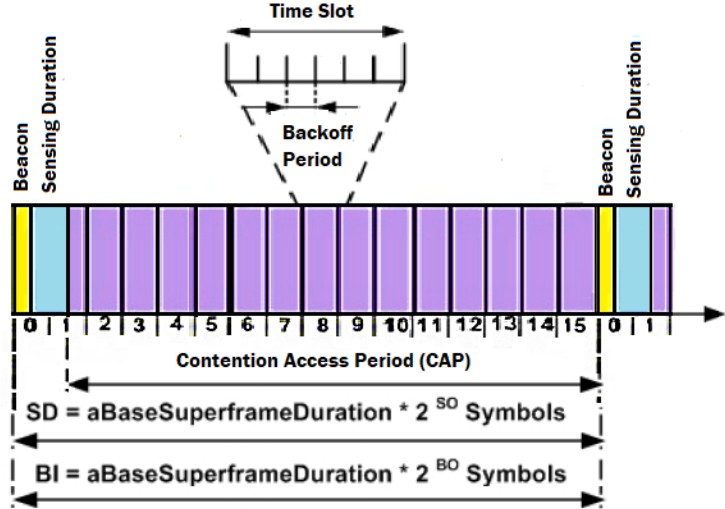

**Figure 10.** Proposed superframe structure and media access control scheme for Internet of Things (IoT) network.

Note that IEEE 802.15.4 devices do adopt CSMA/CA strategy at the MAC level for media access (i.e., manage contention between simultaneous IEEE 802.15.4 transmission). However, the additional sensing phase introduced in this paper is to determine the extended inactive period between the clusters of WLAN frames, which are notably longer. This additional sensing phase serves two purposes: (1) to determine whether media is busy or idle; (2) to determine whether media is idle due to WLAN contention window or due to WLAN extended inactive period. IEEE 802.15.4 devices can transmit data in these extended inactive pockets, which will reduce the overlapping probability of ongoing IEEE 802.15.4 devices transmission with new WLAN communication. In the proposed system model, two types of sensing are performed; the first one is for WLAN active transmission, and second one is for media access control.

After the sensing time, if the medium is idle, the sensor node proceeds with the transmission using CSMA/CA. In case the channel is busy, the IoT nodes use the adaptive wake-up scheduling mechanism explained in the next subsection. Note that CFP in the superframe is not considered in this work since we assume that each IoT sensor node has continuous data to be transmitted to the network coordinator (sink node) and no GTS requests are required to be sent. Thus, the active period consists of CAP only.

### 5.2. Adaptive Wake-Up Scheduling for Sensor Nodes

IoT sensor nodes go into sleep mode for the whole superframe duration if the medium is sensed as busy and wakes up at the start of the next beacon interval. This is done to conserve energy in sensor nodes to extend their lifespan; however, it also results in large end-to-end delay. To overcome this problem, in WAC-MAC, an adaptive wake-up scheduling scheme is proposed based on results of energy detection in which sensor nodes adjust their sleep time if the medium is found busy. This is done through pruning in superframe duration *SD* by half, thus resulting in an early wake-up. The coordinator node, after beacon transmission, immediately starts sensing the spectrum for primary user activity. In the case of the user being present, the coordinator node also adjusts its beacon transmission time accordingly. IoT sensor nodes dynamically adjust their wake-up scheduling based on the result of energy detection to reduce end-to-end-delay, as illustrated in Algorithm 1.

---

**Algorithm 1:** Adaptive Wake-Up Scheduling Algorithm

---

**1** Initialization;
**2** Sensor node wake-up and starts listening the channel;
   **Data:** Input
**3** $E_{WLAN} \leftarrow$ WLAN Energy;
**4** $T_{sleep} \leftarrow$ Sensor Sleep Time;
**5** $T_{tx} \leftarrow$ Transmission Time;
**6** $T_{beacon} \leftarrow$ Beacon Frame;
**7** $T_{sensing} \leftarrow$ Sensing Time;
**8** $SD = 960 \times 2^{SO}$;
**9** $SD = BI \leftarrow$ Superframe Duration = Beacon Interval;
**10** Scan the channel until receive the beacon;
**11** Scan the channel for WLAN activity;
**12** **while** *Scanning for WLAN activity* **do**
**13**    **if** $E_{WLAN} = 1$ **then**
**14**       $BI = \frac{BI}{2}$;
**15**       $T_{sleep} = \frac{SD}{2} \times (T_{beacon} + T_{sensing})$;
**16**       WLAN user is present;
**17**       Superframe duration become half of the current duration;
**18**    **else**
**19**       $BI = BI$;
**20**       $T_{sleep} = SD \times (T_{beacon} + T_{sensing} + T_{tx})$;
**21**       WLAN user is absent;
**22**       Superframe duration remain same for the current duration;
**23**    **end**
**24** **end**
**25** **Output**: $BI$, $T_{sleep}$

---

### 5.3. Adaptive Backoff

For channel access in IEEE 802.15.4, the CSMA/CA algorithm uses only a small range of backoff exponents values in the range of [*macMinBE*, *macMaxBE*], where the value of *macMinBE* by default is equal to 3, and the value of *macMaxBE* is 5. The selection of these backoff exponents by all nodes may result in collisions and cause energy loss in IoT nodes. To solve this issue, we propose an energy-based adaptive backoff scheme where nodes with lower battery time are allocated a lower backoff value, thus giving them a higher priority to transmit. A lower backoff value implies less wait required for channel access. The advantage of this scheme is that nodes with lower energy can complete

their transmissions before their energy gets depleted. Particularly, for energy harvesting-based IoT nodes, this scheme allows nodes to go quickly into sleep mode and harvest energy for future transmissions. Another advantage of this scheme is that nodes are allocated different *macMinBE* values, hence reducing the chance of selecting the same backoff value. Nodes are divided into three priority classes based on their energy level, as shown in the table 1.

**Table 1.** Energy Levels mapping to *macMinBE* Values. BE = backoff exponent.

| Energy Level | Priority Class | MacMinBE Value |
| --- | --- | --- |
| 0%–30% | L1 | 1 |
| 30%–60% | L2 | 2 |
| >60% | L3 | 3 |

The energy level values are calculated based on the residual power of the sensors. Devices that belong to priority class (L1) get their macMinBE value decremented by a value of '2'. Devices that belong to priority class (L2) get their macMinBE value decremented by a value of '1'. Devices that belong to priority class (L3) get their macMinBE value decremented by a value of '0'. Figure 11 depicts the flow chart of proposed CSMA/CA with an adaptive backoff exponent scheme based on the energy level of sensor nodes.

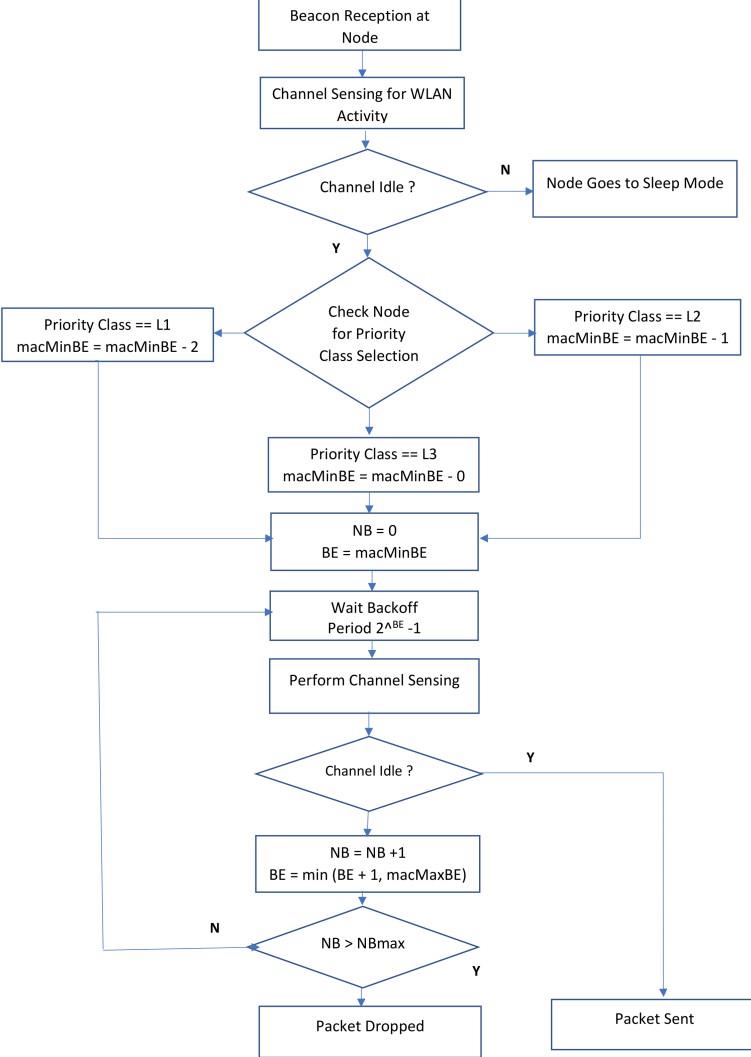

**Figure 11.** Proposed adaptive backoff exponent algorithm for IoT network node. NB = number of backoff.

## 6. Simulation Results

We consider a beacon-enabled star topology IoT network for performance evaluation. A WLAN access point covers an area where IoT nodes are deployed. Both WLAN and IoT nodes are static and operating at 2.4 GHz ISM band. In the proposed model, the considered wireless sensor network (WSN) load is very low; thus contention, congestion, and in-network interference have a minimal impact on WSN, which we aim to loosen in future work. A single interfering WLAN user with a payload size of 512 bytes that resides within transmission range of the Access Point is considered for simulation. Frequency of WLAN transmission is modeled using Poisson distribution. Five active IoT devices having different energy levels from one another are considered. Each IoT node sends uplink data to the network coordinator (sink node) having uplink data of 15 bytes in size. IoT nodes reside within the transmission range of WLAN users, having a radius of 50 m. The distance between the IoT sensor nodes and the coordinator is kept at 15 m. Simulations are performed using MATLAB and IEEE 802.15.4-compliant MicaZ IoT sensor nodes are considered [44]. Parameters for an energy and channel model for an IEEE 802.15.4-compliant IoT device are presented in Table 2. In the simulations, beacons are sent by the coordinator at regular intervals to the sensor nodes. Simulation parameters for WLAN and IoT sensors node are shown in Table 3.

**Table 2.** Energy model and channel model for IoT nodes.

| Parameters Setup for Performance Evaluation | |
| --- | --- |
| **Parameter** | **Value** |
| Path Loss Exponent | 4.0 |
| Noise Power | $-103$ dBm |
| Path Loss Attenuation | $9.98 \times 10^{-5}$ |
| Sensing Time | $0.512 \times 10^{-3}$ |
| Sampling Frequency | 5 MHz |
| Battery | 3 V |
| Sleeping Current | 0.001 mA |
| Idle Current | 0.02 mA |
| Receive Current | 19.7 mA |
| Transmit Current | 17.4 mA |
| Sensing Current | 15.3 mA |

**Table 3.** Simulation parameters for WLAN and IoT nodes. CW = Contention Window; BO = Beacon Order; SO = Superframe Order.

| System Parameters Used for Simulation | | |
| --- | --- | --- |
| **Parameter** | **IEEE 802.15.4** | **IEEE 802.11b/g** |
| Bandwidth | 5 MHz | 22 MHz |
| Transmission Power | 0 dBm | 12 dBm |
| Receiver Sensitivity | $-85$ dBm | $-82$ dBm |
| Transmission Rate | 250 kbps | 54 Mbps |
| Backoff unit time | 320 μs | 20 μs |
| CCA | 128 μs | N/A |
| DIFS | N/A | 50 μs |
| CWmin | N/A | 15 |
| Payload Size | 15 byte | 512 bytes |
| BO = SO | 0 | N/A |
| macMinBE | 3 | N/A |
| macMaxBE | 5 | N/A |
| Max Retransmission | 5 | N/A |

We compare the proposed WAC-MAC protocol with the protocol in the IEEE 802.15.4 standard. Metrics used for comparison include the number of packets received, end-to-end delay, network energy consumption, the energy consumption of individual nodes, and network lifetime.

### 6.1. Number of Packets Received

Figure 12 shows the cumulative number of packets received for the standard and the proposed WAC-MAC protocol. WAC-MAC protocol delivers 780 bytes of data as compared to 700 bytes of data transmitted by the standards scheme. This improvement in the number of packets sent is due to the added function of sensing before frame transmission to identify WLAN inactivity periods, thus resulting in low chances of transmission overlap with the WLAN user.

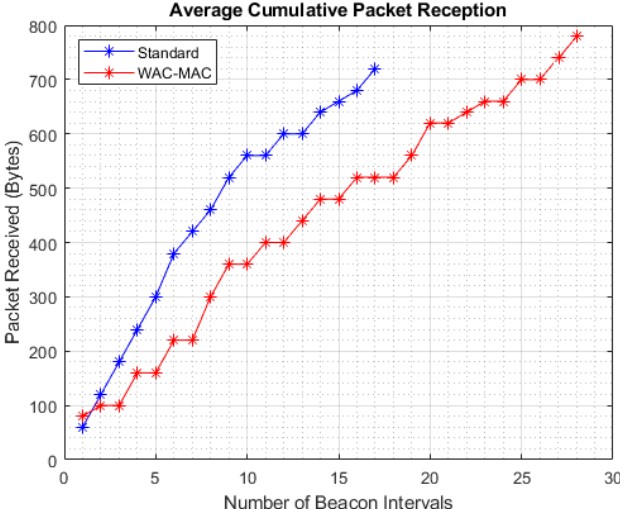

**Figure 12.** Number of packets received. WAC-MAC = WLAN Aware Cognitive Medium Access Control.

### 6.2. End-to-End Delay

Figure 13 shows the average end-to-end delay of packets for both the standard and the proposed WAC-MAC protocols. The standard scheme can achieve lower end-to-end delay as compared to the WAC-MAC protocol. Particularly, for 600 kilobytes of data transmission, the standard scheme takes around 250 ms as compared to 450 ms taken by the WAC-MAC protocol. The lower end-to-end delay of the standard protocol is due to the absence of any sensing mechanism within the frame duration. In comparison, WAC-MAC protocol adds the sensing phase at the start of the frame to detect WLAN users. However, WAC-MAC transmits more data as compared to the standard scheme, at the cost of higher delay.

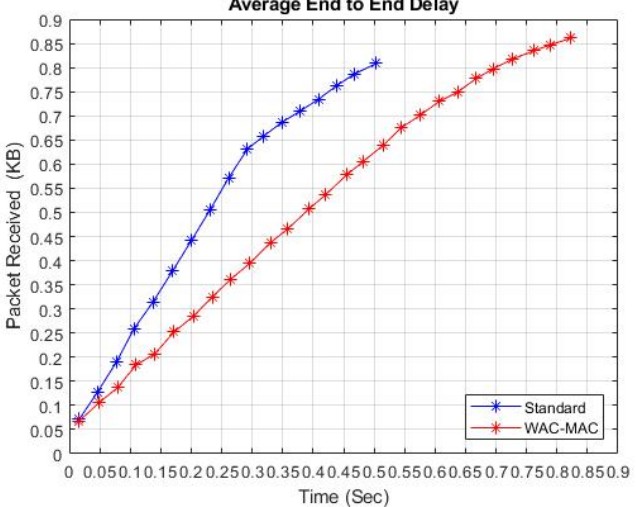

**Figure 13.** End-to-end delay.

### 6.3. Network Energy Consumption

Figure 14 plots network energy consumption, including all the senor nodes, in the standard and the proposed WAC-MAC method. As illustrated in the following figures, the standard protocol consumes 0.25–2 joules more energy as compared to the WAC-MAC protocol. This increased energy consumption is due to packet loss of the sensors caused by the WLAN interference. WAC-MAC reduces simultaneous transmissions between the WLAN users and IoT sensor nodes by adding the sensing function. This reduces packet loss and improves the energy efficiency of the sensor nodes.

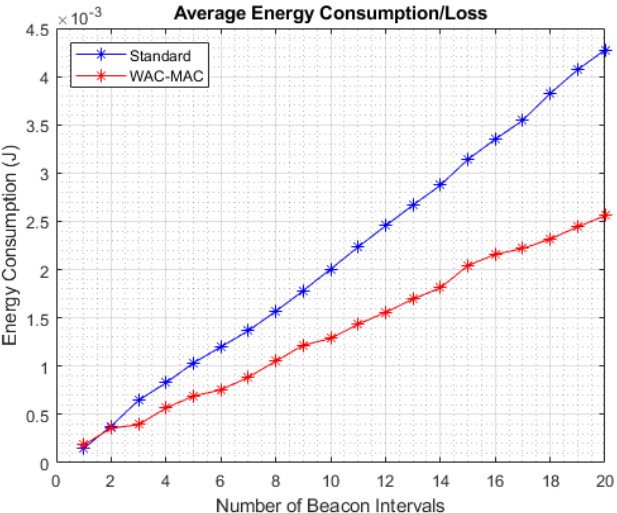

**Figure 14.** Network energy consumption.

### 6.4. Energy Consumption of Individual Nodes

Figure 15 illustrates the energy consumption of individual sensor nodes having different energy levels for the standard and the proposed WAC-MAC protocol. It can be seen that all the sensor nodes improve their energy consumption by 0.5–1 joules using WAC-MAC protocol. The improvement is due to the addition of the sensing phase, as well as the adaptive backoff algorithm, in which nodes with depleted battery life are assigned priority transmissions.

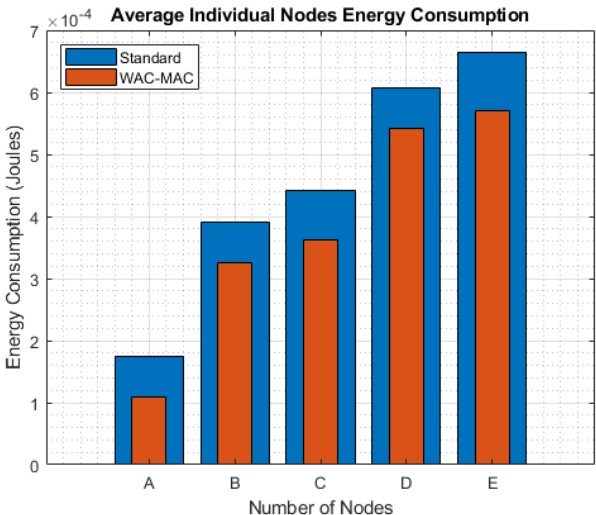

**Figure 15.** Energy consumption of individual nodes.

### 6.5. Network Lifetime

Figure 16 illustrates the lifetime of IoT nodes for the standard and the proposed WAC-MAC protocol. The graph shows the number of beacon intervals required to deplete the energy of a sensor node. For the standard protocol, all nodes deplete their energy within 17 BI. On the other hand, WAC-MAC extends the network lifetime of IoT nodes to 27 BI. This improvement is due to the added function of sensing in the proposed WAC-MAC, as a result of which the sensor nodes transit to the sleep mode if WLAN user is active during the sensing period. This results in the energy conservation of IoT sensor nodes, which would otherwise be wasted due to unsuccessful transmission.

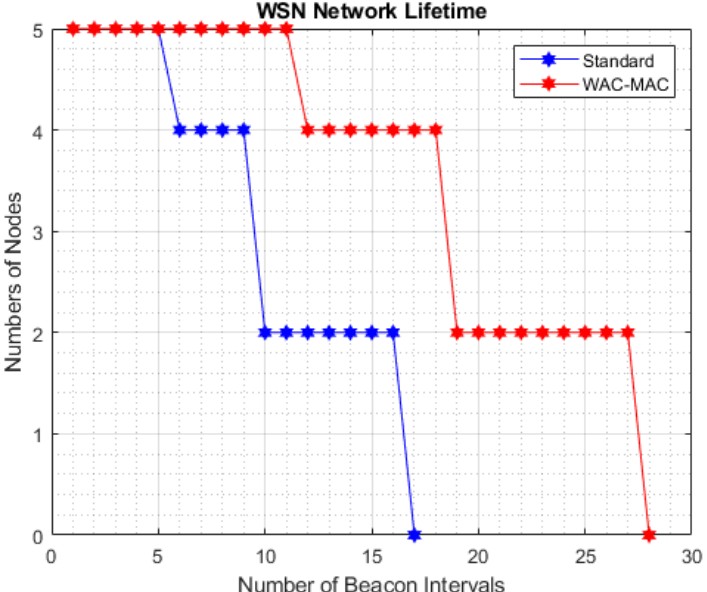

**Figure 16.** Network lifetime.

### 7. Conclusions

In this paper, we consider a co-existence scenario in which low-powered IEEE 802.15.4 IoT devices are subjected to interference caused by high-powered IEEE 802.11 devices. We propose a WLAN Aware Cognitive Medium Access Control (WAC-MAC) protocol that is adopted by the IoT users to reduce the interference of co-existing WLAN users. The proposed WAC-MAC alters the superframe structure of the IEEE 802.15.4 to introduce the sensing phase based on energy detection and incorporates adaptive wake-up scheduling in case the medium is busy. Moreover, an adaptive backoff scheme based on the battery of the sensor node is also included to reduce collisions and improve network lifetime. Results show that the proposed WAC-MAC protocol increases the number of received packets and reduces the energy consumption of IoT nodes.

**Author Contributions:** This article was prepared through collective efforts of all the authors. A.A., M.Z.K., M.A.J. and A.K.B. contributed to the algorithm design and paper writing. A.A. and M.A.J. performed the simulation studies. All authors have read and agreed to the published version of the manuscript.

**Funding:** This research received no external funding.

**Conflicts of Interest:** The authors declare no conflict of interest.

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
