# Peer review of "WLAN Aware Cognitive Medium Access Control Protocol for IoT Applications"

_futureinternet, doi:10.3390/fi12010011_

Round 1

Reviewer 1 Report

This paper introduces an original WLAN-aware Medium Access Control protocol for IEEE802.15.4 devices.
The aim is to prevent IEEE802.15.4 transmissions when a WLAN communication is already ongoing. The basic idea is to make IEEE802.15.4 devices sense the channel before starting any transmissions, so that they can detect possible WLAN signals already on air.

Concerning the paper content, apart from a number of issues that will be detailed in the following, I have one major doubt: IEEE802.11.4 devices do adopt the CSMA/CA strategy at the MAC level. This means that they already sense the channel before starting any transmissions. So, why do the Authors introduce an additional sensing phase in the MAC protocol that they propose? Please, add a clarification.

Analytical remarks:

- page 3, section 3.1. When operating according to the DCF (Distributed Coordination Function) method, IEEE802.11 devices adopt the CSMA/CA strategy at the MAC level. This should be mentioned in this section.

- page 4, section 3.1.1. The whole section refers to the case of WLANs operating according to the DCF method. Please, specify.

- page 4, figure 3. This figure is not very explanatory. For instance, the SIFS interval as well the ACK interval (which are mentioned two lines after the figure) are not represented.

- page 4, line 163. The sentence "IoT devices consist of IEEE802.15.4 compliant sensor nodes ..." should be modified as follows: "The IoT devices considered in this paper consist of IEEE802.15.4 compliant sensor nodes ..."

- page 4, figure 4. The acronym CCA appears here for the first time, and it is not explained. It is introduced only later, at page 6.

- page 5, line 171. Please check the sentence. As far as I know, beacons are sent by the coordinator and not by sink nodes (maybe I missed something).

- page 5, section 3.2.1. The whole section refers to the "Beacon Mode" case. Please, specify.

- page 5, lines 181-182. I think that a figure might be added, which shows the different periods.

- page 7, equation (6). The meaning of the parameter lambda is not explained. Moreover, please check the equation itself.

- page 7, section 4.2, first paragraph. Please check the English.

- page 7, equation (8). In practice (that is, when the algorithm is running on an IEEE802.15.4 device), how is the noise power estimated in order to choose the decision threshold?

- page 8, equation (10). Please, add a reference.

- page 8, equation (12). This equation refers to the total energy consumed during a superframe. Please specify.

- page 9, equation (14). The term in the middle is wrong. I guess that the parameter t_{tx} should be inserted.

- page 9, equation (18). Given (16) and (17), equation (18) could be removed.

- page 9, line 372. Please replace "summer" with "adder".

- page 10, eq. (24) and eq. (25). I don't understand the notation. Which are the meanings of H0 and H1 (that are hypothesis) at the denominators of the two equations?

- page 10, line 288. Please check the English.

- page 10, figure 6. In the figure title, please replace PFA with P_{FA}, which means that FA is a subscript. Do the same in the legend of Figure 7. Moreover, how is the SNR (reported in the x-axis) defined?

- page 11, figure 7. Please, do not use colours to distinguish different curves. Papers are usually printed in black&white.

- page 11, figure 7. As far as I understand, the content of Figure 7 is not consistent with figure 6. In fact, considering in Figure 7 SNR=-15 dB and P_{FA}=0.01, I get a probability of detection around 0.15, while with the same SNR (=-15 dB) and P_{FA}=0.05, I get a probability of detection around 0.35. Therefore, I guess that considering P_{FA}=0.02 (which falls between the two values previously investigated) I should get a probability of detection in the interval [0.15 0.35]. However, Figure 6, which refers exactly to P_{FA}=0.02, shows that for SNR=-15dB the probability of detection is around 0.42. This seems strange, maybe I missed something.

- page 11, figure 8. The content of Figure 8 is not consistent with Figure 7. From Figure 8, which refers to SNR=-10dB, I get a probability of detection of 0.8 when P_{FA} (in the x-axis) is 0.1. However, Figure 7 shows that for the same SNR (=-10 dB) and P_{FA}=0.1, the probability of detection is around 1. Please check.

- page 12, line 323. Check the English.

- page 14, section 6. Please add details on the amount of traffic you considered for the WLAN. How frequent are WLAN transmissions?

-page 15, table 2. Please specify that this table refers to IEEE802.15.4 devices.

-page 16, line 377. Please replace "above figure" with "following figures", as the figure is shown a few lines below.

-page 17, figure 14. Why do different nodes exhibit different energy consumptions?. Please, add a comment.

Bibliography.

The following papers might be added, concerning the use of IEEE802.15.4 in IoT scenario as well as its coexistence with IEEE802.11

Bauwens J., et al., "Coexistence between IEEE802.15.4 and IEEE802.11 through cross-technology signaling," 2017 IEEE Conference on Computer Communications Workshops (INFOCOM WKSHPS), Atlanta, GA, 2017, pp. 529-534.
doi: 10.1109/INFCOMW.2017.8116433

S. Arif and S. H. Supangkat, "Simulation and analysis of ZigBee - WiFi interference," 2014 International Conference on ICT For Smart Society (ICISS), Bandung, 2014, pp. 206-210. doi: 10.1109/ICTSS.2014.7013174

P. Guo, J. Cao, K. Zhang and X. Liu, "Enhancing ZigBee throughput under WiFi interference using real-time adaptive coding," IEEE INFOCOM 2014 - IEEE Conference on Computer Communications, Toronto, ON, 2014, pp. 2858-2866. doi: 10.1109/INFOCOM.2014.6848236

Pasolini, G. et al., Smart City Pilot Projects Using LoRa and IEEE802.15.4 Technologies. Sensors 2018, 18, 1118.

V. Aswin Raaju, J. Mappilllai Meeran, M. Sasidharan and K. Premkumar, "IOT Based Smart Garbage Monitoring System Using ZigBee," 2019 IEEE International Conference on System, Computation, Automation and Networking (ICSCAN), Pondicherry, India, 2019, pp. 1-7. doi: 10.1109/ICSCAN.2019.8878742

A. I. Ali, S. Z. Partal, S. Kepke and H. P. Partal, "ZigBee and LoRa based Wireless Sensors for Smart Environment and IoT Applications," 2019 1st Global Power, Energy and Communication Conference (GPECOM), Nevsehir, Turkey, 2019, pp. 19-23. doi: 10.1109/GPECOM.2019.8778505

Reviewer 2 Report

#Comments:
- Further details can be included for energy consumed during the interference management cycle of the algorithm.

- In Figure 10, are there any retries included when the packet is decided to be dropped. Does it impact performance?

- Congestion is one of the major aspects of access control and during interference, it becomes relatively higher. The authors should also discuss such provisioning in their explanations.

- What are the properties of the standard model used for comparison?

#other comments:
- In (13) and (14) why additional symbols are introduced when the same meaning can be conveyed by a single symbol.

- Figure 9 has some over-labeling. It must be redrawn as it appears to be an overlap.

Reviewer 3 Report

The authors are addressing WLAN Aware cognitive medium access control protocol for IoT applications. Below I am presenting my comments: 1/ The authors must improve the problem in the Abstract. Today, what is missing in the current literature? 2/ Also in the Abstract, present the contributions of the work. 3/ In the 4th and 5th paragraphs of the Introduction section, enhance the problem. In other words, present a brief description of the related work and the problem unsolved yet. 4/ How did you select related work? 5/ Could you provide a comparison Table in section 2? 6/ In Section 2, highlight the main gaps in state of the art today. 7/ Figure 5 is completely strange, and its meaning is unknown. Where is your proposal? What are the architecture and scope? What is yours, and what is a legacy? 8/ Does section 4.1 limit the scope of the article? 9/ In Section 4, you have a collection of Equations. What are yours, and what are not? Do we have novelty in this thematic? 10/ What is the PFA employed in the model, and why? 11/ What is the main section of the article? Section 4 or 5? Put this in the article. 12/ Present input and output data in Algorithm 1. 13/ How did you develop Table 1? 14/ Where is the entry point in the flowchart of Figure 10. 15/ Are your results representative in front of related work? How did you compare your work against similar work? 16/ It s not clear how did you measure the energy consumer option. Explain again. 17/ Enhance the Conclusion with: contributions for society. 18/ What are the main problems of your work? How about fault tolerance, and scalability? 19/ In the conclusion, revisit your idea of cognitive.

Round 2

Reviewer 1 Report

The Authors fulfilled almost all my requests.

However, I have a residual doubt: in their response to my first comments, the Authors state "This additional sensing phase serves two purpose (1) to determine whether media is busy or idle, (2) to determine whether media is idle due to WLAN contention window or due to WLAN extended inactive period. ".

In my opinion, purpose (1) can be achieved by the legacy CSMA/CA strategy natively adopted by IEEE802.15.4 devices.

As far as purpose (2) is concerned, I still do not understand how the additional sensing strategy introduced by the Authors allows discriminating whether the media is idle due to WLAN contention or to WLAN extended inactive period.  Further details are required to clarify this aspect.

Moreover, I do not think that assuming that the Noise Power Spectral Density is equal to -174dBm/Hz, which come from the assumption T=290 K, is accurate. Indeed, no consideration on the antenna noise temperature and the receiver noise figure is carried out. This should be, at least, mentioned in the paper as a (very) simplified approach.

Finally, among the papers that I suggested for the reference list, the Authors did not include the most recent ones. I still suggest their inclusion. 
